# The Clinical Potential of Dimethyltryptamine: Breakthroughs into the Other Side of Mental Illness, Neurodegeneration, and Consciousness

**Frankie A. Colosimo, Philip Borsellino †, Reese I. Krider † , Raul E. Marquez and Thomas A. Vida ***

Kirk Kerkorian School of Medicine at UNLV, 625 Shadow Lane, Las Vegas, NV 89106, USA;
colosf1@unlv.nevada.edu (F.A.C.); borsep1@unlv.nevada.edu (P.B.); krider1@unlv.nevada.edu (R.I.K.);
marqur2@unlv.nevada.edu (R.E.M.)
* Correspondence: thomas.vida@unlv.edu
† These authors contributed equally to this work.

**Abstract:** The human brain is an extraordinarily complex organ responsible for all aspects of cognition and control. Billions of neurons form connections with thousands of other neurons, resulting in trillions of synapses that create a vast and intricate network. This network is subjected to continuous remodeling that adapts to environmental and developmental changes. The resulting neuroplasticity is crucial to both healthy states and many forms of mental illness and neurodegeneration. This narrative review comprehensively examines N,N-dimethyltryptamine (DMT), a naturally occurring hallucinogen and psychedelic compound, focusing on its implications in promoting neuroplasticity via neuritogenesis. We examine DMT's pharmacology, including its interaction with serotonergic, sigma-1, and trace amine-associated receptors and their associated signaling pathways. The therapeutic potential of DMT in both animal models and clinical trials is discussed with impacts on perception, cognition, emotion, and consciousness. We uniquely focus on current directions centered on unveiling the direct mechanisms of DMT's therapeutic effects that demonstrate transformative effects on mental well-being, particularly for conditions like depression, anxiety, and post-traumatic stress disorder. We discuss the connection between DMT and neuroplasticity, offering the potential for forming new neural connections, improving learning, memory, and aiding recovery from brain injuries, including neurorehabilitation and neuroregeneration. The ultimate potential of DMT's therapeutic efficacy to enhance neurogenesis, especially for neurodegenerative conditions, is also discussed.

**Keywords:** N,N-dimethyltryptamine; ayahuasca; psychedelics; neurogenesis; neuritogenesis; neuroplasticity; consciousness; mental illness



## 1. Introduction: DMT's Path to Medical Relevance

Humans have consumed the plant-based brew, ayahuasca, across Central and South America for centuries in ritualistic ceremonies for healing and cultural purposes [1]. Ayahuasca is known for its profoundly spiritual and enlightening effects, inducing an impactful altered state of consciousness. Ayahuasca is produced after boiling bark of the *Banisteriopsis caapi* vine with other additives, most commonly the leaves of the *Psychotria viridis*. Among Amazonian tribes, ayahuasca is considered a gateway for communicating with the gods [2]. N,N-dimethyltryptamine (DMT) is the primary psychoactive compound in this highly revered and potent Amazonian brew. Studies on ayahuasca are well documented with about 1800 papers in the Semantic Scholar database with many excellent reviews [3–5] and systematic reviews [6,7]. Its other active ingredients include several β-carbolines, such as harmine, harmaline, and tetrahydroharmine, which can function as monoamine oxidase (MAO) inhibitors [2]. These molecules are found in the *Banisteriopsis* vine and are suspected to work in conjunction with DMT, primarily by inhibiting its breakdown, to produce the profound psychedelic effects of ayahuasca [8].

DMT is endogenous to plants around the world [9] and even in the brains of mammals [10,11]. DMT, its precursors, and metabolites are detected with liquid chromatography-tandem mass spectrometry in rat brain pineal gland-aCSF (artificial cerebrospinal fluid) microdialysate [12]. However, the role of DMT within the mammalian brain is not well understood. Some of the first studies on DMT in the 1950s spurred hypotheses that it may be implicated in the development of schizophrenia and other psychotic illnesses. This view emerged from its similarities with the positive symptoms of psychosis and the experience of psychedelics, dubbing the molecule a schizotoxin. Further, the transmethylation hypothesis postulated that schizophrenia may result from a stress-induced inborn error of the metabolism that produces a psychoactive compound similar to mescaline [13]. Various investigations studied this idea throughout the 1960s, 1970s, and 1980s with several studies finding higher levels of DMT and other methylated indolealkylamines (from the activity of indolamine N-methyltransferase using tryptophan and serotonin as substrates, see below) in the urine of psychotic patients [14–17]. This prompted debate on whether DMT can act as the "psychomimetic" compound inducing psychotic symptoms in schizophrenic people. However, most of these trials found no significant causal relationship between increased DMT excretion and schizophrenia, suggesting a nonspecific relationship. DMT may be a mediator in the pathway, a simple metabolite, or a biomarker of some other process [16].

In recent years, a psychedelic renaissance using multidisciplinary approaches is rapidly producing a more resolved view of how psychedelic molecules may function [18]. The neuroplastic capabilities of psychedelics are a key feature of their success in treating various psychiatric conditions, especially major depressive disorder [19]. Further, more recent studies suggest that the neuroprotective mechanisms of psychedelics may also have disease-modifying effects in the treatment of neurodegenerative disorders [20]. DMT may prove to be a particularly useful treatment modality for several mental illness conditions and neurodegenerative diseases because of its unique qualities. DMT's short half-life and rapid metabolism within the human body could make it a desirable alternative to other psychedelics whose effects may last several hours. In addition, and in part due to its rapid metabolism, it is not possible to build a tolerance to DMT [21]. People who are treated with DMT would not inevitably require higher doses for therapeutic effects.

This narrative review will provide a comprehensive examination of the pharmacology and therapeutic potential of DMT. Our overall hypothesis is that DMT is an endogenous ligand and may function as a neuromodulator capable of promoting neuroplasticity focusing on neuritogenesis, and possibly neurogenesis, within the human brain. We highlight current clinical trials and a growing body of research that shed light on DMT's potential to offer advanced treatment for patients suffering from many psychiatric and neurodegenerative disorders. These implications span across psychology, neurology, neurorehabilitation, and cognitive science, providing a comprehensive exploration of DMT as a psychoactive molecule.

## 2. DMT Synthesis and Metabolic Transformation: Tryptophan's Path to Hallucinogenic Activity and Oxidative Deamination

DMT possesses a diverse range of interactions within the biological framework. Structurally, DMT's diminutive size (free base, 188.27 g/mol) and hydrophobic nature (logP = 2.573) allow it to cross the blood–brain barrier [22,23]. Its core structure, a tryptamine, is shared with other tryptamines and phenethylamine psychedelics. The compact and unassuming atomic formula of $C_{12}H_{16}N_2$ belies the profound biochemical and behavioral effects that DMT orchestrates in the brain. Equally compelling is the similarity of DMT to psilocin, which contains a single hydroxyl group at position 4 of the indole ring [24]. This similarity in DMT structure to the active form of psilocybin allows for many mechanistic comparisons.

The metabolism of DMT within the body begins with its synthesis. Endogenous DMT is made from tryptophan after decarboxylation transforms it into tryptamine [22,25]. Tryptamine then undergoes transmethylation mediated by indolethylamine-N-methyltransferase (INMT)

with S-adenosyl methionine (SAM) as a substrate, morphing into N-methyltryptamine (NMT) and eventually producing N,N-DMT [26]. Intriguingly, INMT is distributed widely across the body, predominantly in the lungs, thyroid, and adrenal glands, with a dense presence in the anterior horn of the spinal cord. Within the cerebral domain, regions such as the uncus, medulla, amygdala, frontal cortex, fronto-parietal lobe, and temporal lobe exhibit INMT activity, primarily localized in the soma [26]. INMT transcripts are found in specific brain regions, including the cerebral cortex, pineal gland, and choroid plexus, in both rats and humans. Although the rat brain is capable of synthesizing and releasing DMT at concentrations similar to established monoamine neurotransmitters like serotonin [27], the possibility that DMT is an authentic neurotransmitter is still speculative. This issue has been controversial for decades [28] and requires the demonstration of an activity-dependent release (i.e., $Ca^{2+}$-stimulated) of DMT at a synaptic cleft to be fully established in the human brain.

The chemical synthesis of DMT in vitro is remarkably simple. The most popular method, known for its accessibility, involves the reductive amination of tryptamine in aquatic formic acid and a sodium cyanoborohydride milieu, achieving approximately 70% purity in a single step [29]. Alternatively, a more complex route begins with the acylation of indole with oxalyl chloride at position 3, followed by the addition of dimethylamine, and subsequent reduction with lithium aluminum hydride to derive DMT [30]. Post-synthesis extraction typically employs a basic aqueous workup into an organic solvent like chloroform, with purification achieved either as a free base through sublimation or as a salt after crystallization [25].

After endogenous synthesis, DMT's activity through the body is believed to involve three steps: an initial crossing of the blood–brain barrier via $Mg^{2+}$ and ATP-dependent uptake, followed by uptake through serotonin uptake transporters (SERT) on neuronal plasma membranes, and finally, sequestration into synaptic vesicles from the cytoplasm via the vesicular monoamine transporter 2, VMAT2 [31]. Remarkably, DMT demonstrates strong bindings to SERT and VMAT2, with binding-to-uptake ratios of >11 and >10, respectively [22,31].

The pharmacodynamic landscape of DMT is equally interesting. Unlike many substances, DMT does not exhibit tolerance upon repeated administration in several animal models [32,33], or in humans [21,34]. However, mild cross-tolerance occurs in humans tolerant to lysergic acid diethylamide (LSD) [35,36]. Tolerance appears to be more inclined toward peripheral effects rather than subjective experiences, with partial tolerance in rats reported within dose ranges of 3.2–10 mg/kg every 2 h for 21 days [33].

When exogenous DMT formulations are introduced into the body, whether through ingestion, intravenous injection, or inhalation, they undergo rapid absorption into the bloodstream. DMT can access cerebral circulation in as little as 10 s after the intravenous injection of radiolabeled DMT into rabbits [22,25,37–40]. Once DMT is absorbed, nearly all of it undergoes full metabolism and clearance from the body with first-pass metabolism in the liver, a process known as N,N-dimethyltryptamine-N-oxidation. Monoamine oxidase A (MAO-A) is uniquely responsible for converting DMT into indoleacetic acid, an inactive metabolite and the primary product of MAO-A degradation, along with other secondary metabolites, including the psychoactive compound, N,N-dimethyltryptamine-N-oxide (DMT-NO) [41]. Twenty-four hours after initial administration, only about 0.16% of unaltered DMT is recovered in urine [42].

The role of MAO-A in DMT metabolism has been further elucidated using MAO inhibitors (MAOI) in experimental subjects. When administered alongside DMT, a noticeable shift occurs from indoleacetic acid to favoring DMT-NO production. When DMT is administered orally, it undergoes rapid peripheral metabolism, resulting in insufficient levels in the bloodstream to penetrate the brain and induce its full psychedelic effects. However, this limitation can be overcome by co-administering MAOI, allowing DMT to partially bypass first-pass metabolism in the liver [43].

In contrast, when DMT is inhaled, it is primarily metabolized in the liver through a CYP-dependent process, which is markedly less efficient than the MAO-dependent process [43]. Consequently, subjects who inhale DMT exhibit higher levels of unmodified DMT, and lower levels of inactive metabolites compared to those who consume it through other routes. This disparity in metabolite levels helps explain the significant differences in the psychological experiences reported by subjects after smoking or injecting DMT compared to ingesting it [43].

## 3. DMT's Multi-Receptor Pharmacodynamics: Serotonergic, Sigma-1, and Trace Amine-Associated Pathways

### 3.1. Serotonin Receptors

The effects of DMT, and other psychedelics, are actuated in large part through the binding of serotonin (5-HT) receptors. DMT is a partial agonist to several serotonin receptors, including $5\text{-HT}_{1A}$, $5\text{-HT}_{1B}$, $5\text{-HT}_{1D}$, $5\text{-HT}_{2A}$, $5\text{-HT}_{2B}$, $5\text{-HT}_{2C}$, $5\text{-HT}_{5A}$, $5\text{-HT}_6$, and $5\text{-HT}_7$, with varying affinities [38] (Table 1). Although primary receptors are thought to be the 1A, 2A, and 2C subtypes, the $5\text{-HT}_{2A}$ receptor appears to be the predominant mediator of the hallucinogenic properties for most psychedelics, including DMT [25,44]. The $5\text{-HT}_2$ receptor family comprises $G_{q/11}$-mediated receptors that primarily utilize phospholipase C as well as phospholipase A2 s messengers [25]. DMT induces the head-twitch response, which is an indicator of the $5\text{-HT}_{2A}$ activity of psychedelics in rodents [38]. Interestingly, DMT has been found to induce fewer head-twitch responses than other psychedelics [25].

Evidence for serotonin receptors in psychedelic mechanisms of action is robust; however, it is incomplete in explaining the complete psychedelic experience [45,46]. For example, DMT also interacts with a variety of ionotropic and metabotropic glutamate receptors, which may play a synergistic role in some of the sensorial effects of DMT. More recent studies demonstrate an interaction between DMT and other receptors, notably the sigma-1 receptor and trace amine-associated receptors (TAARs).

### 3.2. Sigma-1 Receptor

Sigma-1 receptors were previously believed to belong to the opioid receptor family [47]. However, they were classified as orphan receptors located throughout the CNS and the rest of the body [48]. Sigma-1 receptors are primarily located at the interface between the endoplasmic reticulum (ER) and mitochondrion, referred to as the mitochondria-associated ER membrane [47,49,50], and high concentrations of receptor ligands can translocate them to the plasma membrane [46]. From here, they function as molecular chaperones that modulate various voltage-gated ion channels [46,51]. DMT is the only known endogenous mammalian N,N-dimethylated trace amine to bind the sigma-1 receptor [51], though with relatively low affinity. DMT administration causes an inhibition of $Na^+$ ion channels in cardiomyocytes and rat hypermobility, suggesting its role as a sigma-1 ligand [51]. The sigma-1 receptor is implicated in having a neuroprotective role against a myriad of neuropsychological conditions via neuroplastic and anti-inflammatory processes [52]. Thus, the sigma-1 receptor may be another potential mediator of DMT's psychogenic effects and a target in the treatment of neurodegenerative diseases (discussed below).

### 3.3. Trace Amine-Associated Receptors (TAARs)

TAARs are a group of G protein-coupled receptors found throughout the mammalian nervous system, the effects of which are mediated via adenylate cyclase and the increased production of cAMP. They contain binding sites of trace amines, supporting their role as neurotransmitters/neuromodulators within mammalian brains [53]. DMT binds to TAARs with high affinity [54], suggesting its possible role as a neuromodulator. The role of TAARs in the function of psychedelics is not well understood. One hypothesis suggests that TAARs may play a role in the visual/sensory experience of psychedelics [45], which is not well explained with $5\text{-HT}_{2A}$ receptor activation alone.

**Table 1.** DMT Receptor interactions and effects of receptor activation.

| Receptor | Affinity: $K_i$ (nM) | DMT Action | Signaling Pathway(s) | Downstream Effects | Reference(s) |
|---|---|---|---|---|---|
| 5-HT$_{1D}$ | 39 | agonist | $G_i/G_0$ | inhibits neurotransmission | [22,37–40] |
| 5-HT$_{2A}$ | 127 | agonist | $G_q$ | increases phosphoinositide hydrolysis; increases mTOR-dependent structural plasticity, neurite growth, and spinogenesis | [22,55,56] |
| 5-HT$_{1B}$ | 129 | agonist | $G_i/G_0$ | inhibits neurotransmission | [22,37] |
| 5-HT$_{1A}$ | 183 | agonist | $G_i/G_0$ | acute inhibition of dorsal raphe firing; anxiolytic, and antidepressant | [22] |
| 5-HT$_{2B}$ | 184 | agonist | $G_q$ and β-arrestin 2 | transport and regulation of serotonin plasma levels; vasoconstriction and platelet morphology; maintains cardiac valve leaflets | [22,57–59] |
| 5-HT$_7$ | 206 | partial agonist | adenylate cyclase; CDK5, and GTPase Cdc42 | serotonergic system-related neuropsychiatric disorders; prolonged activation of dendritic spine formation and synaptogenesis in cortical and striatal neurons; acute activation of neurite elongation in striatal and cortical neurons; establishment of correct neuronal cytoarchitecture during development and remodeling of neuronal circuits in the mature brain | [22,38–40,60–62] |
| 5-HT$_{2C}$ | 360 | agonist | $G_q$ | desensitizes over time, unlike 5-HT$_{2A}$ | [22,63] |
| 5-HT$_6$ | 464 | partial agonist | $G_s$, Erk1/2, Jun, mTOR | modulation of cognitive processes, mood regulation, and motivated behaviors | [22,38–40,64,65] |
| 5-HT$_{1E}$ | 517 | agonist | $G_i/G_0$ | inhibits neurotransmission | [22,37] |
| 5-HT5$_A$ | 2135 | partial agonist | $G_i$ and $G_0$ | control of circadian rhythms, mood, and cognitive function, and implicated in schizophrenia | [22,66,67] |
| Sigma-1 | $K_d$ ~15 μM | agonist | BDNF and EGF | neural plasticity, protection from oxidative stress, and antidepressant | [22,51,68] |
| TAAR1 | Unknown | agonist | $G_s$ | dopamine efflux via dopamine transporter internalization | [22,54,69] |

## 4. Elucidating the Molecular Basis of DMT's Role in Neuronal Development: Insights into Neurogenesis and Neuritogenesis Mechanisms

Psychedelics have pronounced neuroplastic effects, which are best described as neuritogenesis. For example, psilocybin causes rapid, lasting changes in dendrite architecture in vivo after a single dose in mice [22,70]. Psilocybin also leads to lasting synaptogenesis within the hippocampal SV2A region of porcine brains up to only 7 days post intervention, which is most likely neuritogenesis but the studies looked at the level of brain tissue [71]. Neuritogenesis is also observed with other serotonergic psychedelics such as 2,5-dimethoxy-4-iodoamphetamine (DOI), a commonly used psychedelic for the study of 5-HT$_{2A}$ receptor

activation. The treatment of mice with DOI leads to enhanced synaptic plasticity within pyramidal neurons of the frontal cortex [72]. DOI and lysergic acid diethylamide (LSD) can quickly lead to neuritogenesis in vivo and in vitro [56].

This neuritogenic effect is also observed with DMT in vivo. When treated with 10 mg/kg of DMT, rats showed increased density in the dendritic spines within cortical pyramidal neurons in the PFC 24 hours post treatment [56]. Even at sub-hallucinogenic doses, similar effects were observed [56]. This neuritogenesis is believed to be mediated through a 5-HT$_{2A}$ receptor signaling pathway given that concurrent treatment with the 5-HT$_{2A}$ receptor antagonist, ketanserin, led to an inhibition of the neuritogenic effects. The 5-HT$_{2A}$ receptor has a diverse array of downstream mediators that could be responsible for the differences in subjective and psychological effects [73]. The main downstream mediator of interest is the mTOR protein, which plays a major role in cell growth and translation of proteins required for synaptogenesis [74]. In support of this postulate, pretreatment with rapamycin completely blocked the neuritogenic effect of 5-HT$_{2A}$ receptor agonism [56].

The link between the 5-HT$_{2A}$ receptor and mTOR modulation is not fully understood, suggesting that additional receptor sites may be responsible for the neuritogenic effects of psychedelics. As mentioned earlier, DMT has the ability to modulate the sigma-1 receptor [51]. Sigma-1 receptors are intracellular chaperones that regulate ER-mitochondrial calcium signaling. They play an important role in dendritic arborization affecting cellular excitability and long-term potentiation through their effects on ion channels [49]. Sigma-1 receptors also play a role in potentiating not only neuritogenesis but also neurogenesis. Behavioral studies in mouse models show that DMT treatment leads to neuronal proliferation in the dentate gyrus of the hippocampus. This includes the proliferation of neural stem cells, the migration of neuroblasts, and the formation of new neurons within the hippocampus. These cellular changes are also functional, as mice treated with DMT performed better on spatial learning and memory tasks. Importantly, the antagonism of the sigma-1 receptor led to the inhibition of these effects [75]. While these results are promising in providing evidence for the roles of DMT and sigma-1 receptors in stimulating neurogenesis, other studies have revealed the opposite effect. In mice treated with Aβ1-42 oligomers to induce Alzheimer's disease (AD)-like deficits, DMT had a negative effect on neurogenesis compared to PRE084, a selective sigma-1 agonist. The concurrent binding of DMT to both the 5-HT$_{2A}$ and sigma-1 receptors with varying affinities may have influenced this negative effect on neurogenesis, which could also be related to the disease model used in this study [76].

In addition to the sigma-1 receptor, synaptogenic responses occur after agonism at additional receptor sites including 5-HT$_6$ and 5-HT$_7$. Since DMT can bind to these receptors, we speculate that its mechanism of action may include their diverse effects after activation. For example, the agonism of 5-HT$_7$ facilitates synaptogenesis and increases the density of dendritic spines in the forebrains of mice. Chronic activation leads to sustained effects within the postnatal cortical and striatal neurons. Acute activation resulted in neurite elongation within these regions [61]. This effect also occurs within the hippocampus at early postnatal stages [77]. The 5-HT$_7$ receptor also mediates arborization in rat forebrains due to agonism in late adolescent development, and is maintained into adulthood [78]. This effect may occur in an age-dependent manner, with the structural effects being attenuated into adulthood [77]. Another target of interest includes the 5-HT$_6$ receptor, which plays a role in the development of neural circuits. Specifically, its activation is implicated in neuronal migration and neurite outgrowth [79]. Constitutive activation of 5-HT$_6$ through its interaction with Cdk5 leads to neurite growth, which was abolished through the administration of a 5-HT$_6$ receptor antagonist [80]. These 5-HT$_6$ receptor sites have received less attention as potential mediators of psychedelic-mediated neuritogenesis, and future investigation into their contribution is certainly warranted based on the preliminary findings of in vitro and in vivo studies. Experiments testing the role of DMT in 5-HT$_6$ and 5-HT$_7$ receptor interactions are predicted to yield mechanistic insights.

The role of brain-derived neurotrophic factor (BDNF) in the neurotrophic response to psychedelics is a complex and evolving area of research, with various studies shedding light on its involvement. BDNF stimulates the growth of 5-HT neurons, and 5-HT affects the expressions of genes that control for BDNF synthesis [81]. BDNF has long been studied as a central mediator of synaptic plasticity. It plays a role in the growth and development of GABAergic and glutamatergic synapses and plays a role in long-term potentiation and memory [82]. When BDNF is released into the extracellular space, it activates α-amino-3-hydroxy-5-methyl-4-isoxazolepropionic acid receptors (AMPARs) at the post-synaptic membrane and also interacts with the TrkB receptor in an autocrine fashion [83].The serotonergic psychedelic, DOI, which also acts at the 5-HT$_{2A}$ receptor, leads to the upregulation of BDNF in the frontal cortex [84]. Further, after DOI treatment, the levels of BDNF protein also increased in a rat cortical neuron model of psychedelic action [56]. A similar increase in BDNF transcripts occurs after DOI stimulation, with a possible contribution of cAMP response element binding (CREB) protein interaction [85]. Finally, sigma-1 receptor activation modulates BDNF [86,87], which suggests DMT may also be implicated. However, LSD and psilocin bind directly to the tropomyosin receptor kinase B (TrkB) receptor, and it is possible that this may be ultimately responsible for the downstream mediation of the mTOR pathway. In support of this idea, the treatment of mice with a TrkB receptor antagonist, ANA-12, leads to the extinction of both the neurogenic effects of BDNF and DOI [56,88]. Further, BDNF levels in the blood increase after administration with LSD, suggesting that psychedelics trigger neurotrophic signaling cascades [89–91]. Curiously, DMT does not increase the levels of BDNF within human participants even at high doses [92]. However, increased BDNF levels are not consistent in experiments with LSD or psilocybin [93]. More studies are needed to definitively determine if DMT administration affects plasma BDNF levels. Ultimately, the connection between BDNF, TrkB, and psychedelics remains to be fully elucidated, though it is likely that the same downstream mediators are similarly implicated in neuritogenesis.

TrkB receptor stimulation primarily activates three intracellular pathways including the phospholipase C (PLC)-γ pathway [94], the mitogen-activated protein kinase/extracellular signal-regulated kinase (MAPK/Erk) pathway [95], and the phosphoinositide-3-kinase–protein kinase B (PI3K-Akt) pathway [96]. The PI3K-Akt pathway is directly implicated as one of the main drivers of BDNF's ability to interact with mTOR and lead to the upregulation of protein synthesis [83]. BDNF treatment leads to the increased phosphorylation of mTOR, which is mediated through TrkB activation. mTOR activation then leads to the phosphorylation of factors 4EBP, ribosomal protein S6, and p70S6K, which leads to the cap-dependent translation of proteins involved in synaptic plasticity [97]. Additionally, the MAPK/Erk pathway leads to the activation of mTOR as well as the CREB and the elongation factor 4E (elF4E). These various downstream mediators drive synaptic plasticity. BDNF also plays a role in modulating synapse strength through the activity of the TrkB-PSD-95 complex and plays an important role in modulating long-term synaptic depression through the activation of p75NTR [83].

Overall, the mechanism of DMT's neuritogenesis involves complex intracellular signaling models that are likely to coalesce, producing the observed neuritogenic effects (Figure 1). The main mediator is currently thought to be related to the modulation of mTOR, likely through the downstream actions of 5-HT$_{2A}$ agonism and the modulation of TrkB. Other potential contributory sites of action include the sigma-1, 5-HT$_6$, and 5-HT$_7$ receptor sites. More studies are warranted to determine the magnitude of their respective effects.

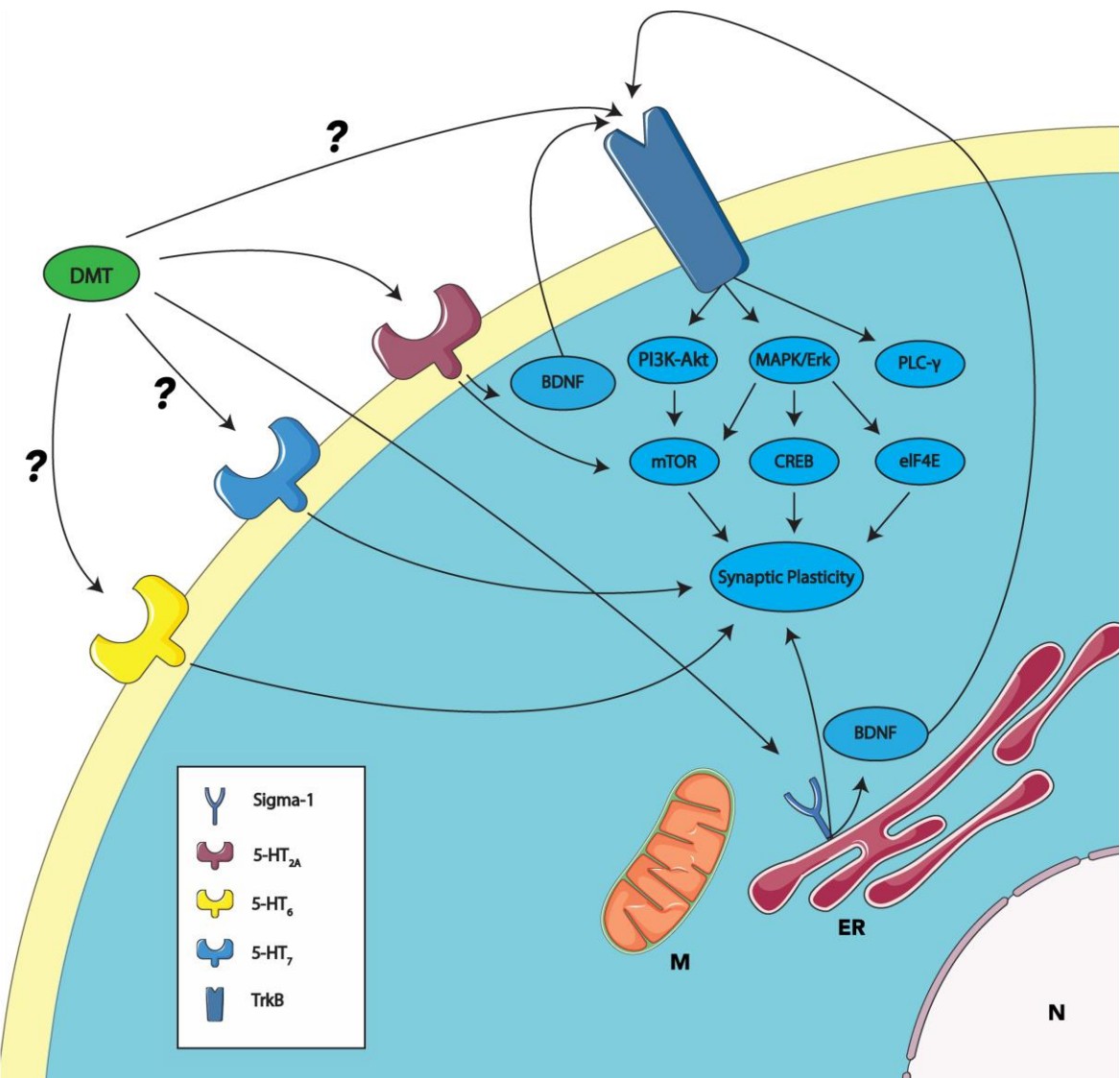

**Figure 1. Proposed mechanisms of DMT-induced synaptic plasticity through receptor interactions**. This model schematically represents the putative actions of DMT on synaptic plasticity through the established activation of the sigma-1 and 5-HT$_{2A}$ receptors (as described in Section 4; see also [63,75,98–100]). The DMT activation of the sigma-1 receptor, which is implicated in the modulation of synaptic plasticity, may enhance BDNF's activity [82]. BDNF, acting in an autocrine manner upon the TrkB receptor, is proposed to elevate the PI3K-Akt, MAPK/Erk, and PLC-gamma pathways, culminating in the upregulation of synaptic plasticity. The PI3K-Akt and MAPK/Erk pathways are responsible for activating mTOR, a pivotal mediator of synaptic plasticity. The MAPK/Erk pathway also contributes to the activation of downstream elements like CREB and eIF4E, which are instrumental in mediating synaptic plasticity. DMT's interaction with the 5-HT$_{2A}$ receptor is postulated to influence the mTOR pathway and, consequently, synaptic plasticity. DMT is predicted to bind with the 5-HT$_6$ and 5-HT$_7$ receptors (indicated with a **?**). The activation of these receptors is also linked to synaptic plasticity. Emerging evidence suggests the possibility of psychedelics, including DMT, binding directly to the TrkB receptor (indicated with a **?**), initiating the signal transduction processes outlined above. ER, endoplasmic reticulum; M, mitochondria; and N, nucleus. Some graphic components from Servier Medical Art were used to draw parts of this model. Servier Medical Art by Servier is licensed under a Creative Commons Attribution 3.0 Unported License (https://creativecommons.org/licenses/by/3.0/, accessed on 30 October 2023).

## 5. Therapeutic Insights for DMT: Bridging Animal Models and Clinical Studies in Humans

*5.1. DMT and Analogs: Neurobehavioral Insights from Rodent Studies*

The potential therapeutic properties and physiological effects of DMT and its analogs have been investigated across various rodent studies, shedding light on its interactions with neurological and behavioral parameters. These studies use either ayahuasca, pure DMT, or pure 5-methoxy-N,N-dimethyltryptamine (5-MeO-DMT), which is structurally homologous to DMT and exhibits similar psychedelic qualities [101]. High doses of DMT significantly inhibit aggressive behaviors in rats via shock-elicited fighting within a metallic cage with similar trends observed for 5-MeO-DMT, albeit at lower doses [102]. This is compared with the biphasic response of LSD, highlighting the differential behavioral impacts between LSD, DMT, and 5-MeO-DMT. Some animal models of anxiety behaviors have used an X-maze along with monitoring for 5-HT syndrome behaviors and a swift back-and-forth motion of the body, "wet dog shakes" [103,104]. Among various 5-HT agonists, 5-MeO-DMT manifested a threshold dose of 2.5 mg/kg for both anxiety behaviors and 5-HT syndrome behaviors, with a halved entry ratio observed at a mean of 0.24 mg/kg. Similarly, intrathecal administration of 5-HT and 5-HT agonists, particularly with 5-MeO-DMT, induced wet dog shakes, implicating that action on 5-HT$_2$ receptors is possibly located in the spine, and eliciting novel motor behaviors [105].

Ayahuasca has therapeutic potential in reducing the rewarding effects of alcohol use disorder (AUD). In a series of experiments on mice using different doses of ayahuasca and *B. caapi* and *P. viridi* extracts, the ability to alter conditioned place preference (CPP) was tested [106]. An intermediate dose of ayahuasca led to mice preferring the environment associated with that dose, suggesting that ayahuasca, at a specific dosage, had a rewarding effect on the mice. Ayahuasca, when given before ethanol, blocked the development of a preference for ethanol-associated environments. However, *B. caapi* and *P. viridi* extracts did not produce these same effects. This suggests that the combination of *B. caapi* and *P. viridi* extracts in ayahuasca appears to have unique therapeutic potential. These results are interpreted to reflect the primary activity of DMT interacting with 5-HT$_{2A}$ and 5-HT$_{2C}$ receptors as an agonist. In addition, other components of ayahuasca like tetrahydroharmine, acting as a serotonin reuptake inhibitor, and harmine also bind to 5-HT$_{2A}$ and 5-HT$_{2C}$ receptors [106]. In evaluating a more unique application of treating addiction and AUD, mice were administered alcohol, followed by a 5-HT$_{2A}$ antagonist (M100907) or placebo, followed by ayahuasca or placebo to evaluate alterations in ethanol self-administration. A notable reduction in ethanol preference and consumption was observed after ayahuasca administration, an effect attenuated by pretreatment with a 5-HT$_{2A}$ antagonist. [107]. This suggests a promising role for ayahuasca, and by extension DMT, in managing AUD.

DMT's effects on anxiety in rats were tested after a 10 mg/kg dose, and behavioral assessments were conducted an hour post administration to circumvent the confounds of acute serotonin syndrome [108]. Notably, the findings reveal acute anxiogenic effects following DMT administration, suggesting a complex interplay in its effects on anxiety. Interestingly, DMT has no significant impact on fear conditioning, but it exhibits enhanced effects on fear-extinction training when administered an hour before the training session. This observation hints at a potential therapeutic use like 3,4-methylenedioxymethamphetamine (MDMA) in facilitating fear extinction training, which could be beneficial in the treatment of post-traumatic stress disorder (PTSD, see below). Further, the study revealed that DMT's effects in forced swimming tests were nearly indistinguishable from those of ketamine, indicating potential acute antidepressant properties. This finding raises possibilities for DMT's role as a rapid-acting antidepressant agent. Additionally, the inclusion of harmine was proposed to potentially offer a synergistic or additive effect alongside DMT [108]. Additionally, ayahuasca administered to fear-conditioned rats showed impairment in fear memory reconsolidation rather than fear extinction, with no anxiolytic effects noted [109]. Further studies are needed to investigate the therapeutic prospects of DMT and ayahuasca in addressing conditions such as alcohol use disorder (AUD), anxiety, and depressive disorders.

Importantly, in future studies, a shift from ayahuasca to using pure DMT or 5-MeO-DMT will avoid ayahuasca's polypharmacology, which complicates precise mechanistic analyses and the interpretation of results.

*5.2. Exploring the Effects and Safety of DMT in Healthy Subjects through Clinical Trials*

In recent years, clinical trials on healthy humans have emerged, exploring the effects and safety profiles of DMT administration. These are international in scope and set the stage for trials to assess therapeutic benefits. A phase 1 double-blind randomized control trial (RCT) is currently in the recruiting stage (NCT05559931, Algernon Pharma, Leiden, Netherlands) [110]. The study design involves an intravenous infusion of DMT over six hours, initiated with a 1.5 mg loading bolus, followed by a 0.105 mg/min infusion, with this regimen repeated across six total doses on Days 1, 3, 5, 8, 10, and 12 of a two-week treatment period. A placebo group is also included to ensure a robust analysis. The primary outcomes focus on overall safety, encompassing parameters such as ECG, vital signs, physical examination findings, lab values, injection site reactions, the effect on the Columbia-Suicide Severity Rating Scale (C-SSRS), occurrences of psychotic symptoms, 5-HT toxicity syndrome, and general adverse events [110].

Several clinical trials are focused on investigating the acute dose-dependent effects of DMT in healthy subjects at the University Hospital, Basel, Switzerland. These studies aim to better understand the impact of DMT on altered states of consciousness and subjective experiences. One is a phase 1, placebo-controlled, double-blind crossover study involving five active substance conditions with varying DMT doses (ranging from 0.6 mg/min to 2.4 mg/min) and a placebo. Additionally, a patient-guided titration session will be conducted. The trial measures altered states of consciousness profiles (5D-ASC) and subjective effect ratings over 12 months to assess the acute effects of DMT (NCT05384678). [111]. A related trial explores the acute dose-dependent effects of DMT through bolus administrations. This phase 1 trial involves randomized subjects receiving intravenous DMT in different bolus doses or a placebo. The study measures altered states of consciousness profiles (OAV) and subjective effect ratings over time. It aims to evaluate the effects of DMT in a controlled environment (NCT05695495) [112]. Finally, a double-blind, placebo-controlled, crossover study is underway with 27 healthy participants and four different DMT doses. The trial evaluates altered states of consciousness, subjective effect ratings, mood ratings, mystical-type experiences, autonomic effects, plasma levels, oxytocin levels, renal clearance, and adverse effects to gain comprehensive insights into DMT's effects. The study's results show that bolus doses of DMT quickly induced intense psychedelic effects, while DMT infusions led to gradually increasing, dose-dependent effects peaking at about 30 min. Bolus doses were associated with more negative effects and anxiety. Interestingly, all DMT effects rapidly disappeared within 15 min after stopping the infusion, with a short early elimination half-life of 5.0–5.8 min and a longer late elimination phase of 14–16 min. Despite rising plasma concentrations, subjective effects remained stable from 30 to 90 min, suggesting acute tolerance to continuous DMT administration. Overall, intravenous DMT, particularly via infusion, is promising for the controlled induction of tailored psychedelic states, potentially benefiting clinical and therapeutic contexts (NCT04353024, [92].

Trials have also tested the effects of inhalation as a mode of DMT administration. A phase 1 double-blind RCT investigates the acute and subacute effects of inhaled DMT, administered in a 60 mg experimental dose during a two-hour session, contrasting with a 1 mg placebo dose (NCT05901012, Universidade Federal do Rio Grande do Norte, Natal, Rio Grande do Norte, Brazil; estimated completion date of August 2023) [113]. Interestingly, prior experience with DMT is seemingly a part of the inclusion criteria, and a vaporizer device is utilized for administration. The primary outcome measures include blood pressure, heart rate, respiratory rate, and oxygen saturation, assessed seven times throughout the session. Another phase 1 trial, a non-randomized sequential assignment open-label study enrolled 27 participants with inclusion criteria of prior DMT experience and proof of COVID vaccination (NCT05573568, results submitted in May 2023; Biomind Labs Inc.

Toronto, Ontario, Canada and Universidade Federal do Rio Grande do Norte, Natal, Rio Grande do Norte, Brazil) [114]. The trial comprised two sessions of inhaled DMT, with a low dose administered initially, followed by a higher dose after a two-hour interval. The dosages varied among five groups, ranging from 5 mg to 15 mg in the first session, and 20 mg to 60 mg in the highest dosage group. The primary outcomes included the monitoring of clinical and psychiatric adverse events up to a month post dosage, and assessments of blood pressure, heart rate, respiratory rate, and oxygen saturation changes [114]. Although distinct in its outcomes, another trial tested vaporized 5-MeO-DMT on the psychedelic experience using various questionnaires such as the novel peak experience scale (PES), the mystical experience questionnaire (MEQ), the ego dissolution inventory (EDI), the challenging experience questionnaire (CEQ), and the five-dimensional altered states of consciousness questionnaire (5D-ASC). The safety profile measured with vital signs was excellent with no major changes at 1 and 3 h. 5-MeO-DMT demonstrated a dose-dependent increase in the intensity of the psychedelic experience as indicated by various questionnaires, except for the CEQ. Notably, significant effects were observed when individuals received single doses of 6, 12, and 18 mg, particularly in terms of PES and MEQ ratings. The most pronounced effects across multiple rating scales, including PES, MEQ, EDI, and 5D-ASC, were observed when participants underwent individualized dose escalation with 5-MeO-DMT. However, measures related to cognition, mood, and overall well-being remained unaffected [115]. 5-MeO-DMT is associated with ego dissolution [116], while DMT is known for its ability to produce profound and complex visual imagery during the psychedelic experience [117].

Two studies stand out as they investigate prosocial emotional processing and the connection between mindfulness and psychedelics. The first is a completed phase 1 study using pharmacological electroencephalography (EEG), behavioral outcome measures, and biomarker detection. It investigated the effects of DMT and harmine, exploring the neurodynamics on social cognition, prosocial emotional processing, and self-referential processing in healthy subjects. (NCT04716335, Psychiatric University Hospital, Zurich, Switzerland) [118]. The second trial is an early phase 1 study, using functional magnetic resonance imaging (fMRI) to assess brain connectivity before and after a meditation group retreat. Participants were administered either ayahuasca or a placebo to explore the nexus between mindfulness and psychedelics. The primary outcomes were focused on the fMRI comparisons of brain connectivity at rest or during meditation among experienced meditators, and the effects of DMT-enhanced mindfulness on the default mode network using seed-based variance analysis (SVA) and independent component analysis (ICA). Completed on 15 September 2023, however, the study's results are yet to be published or posted (NCT05780216, Psychiatric University Hospital, Zurich, Switzerland) [119].

These diverse studies that underscore the burgeoning interest and rigorous investigation into the therapeutic potential and safety profiles of DMT are summarized in Table 2. Common themes in the early trials we have outlined are DMT's rapid duration of effects, a lack of induced tolerance after repeated administrations, and potent psychedelic experiences upon administration.

**Table 2.** Clinical trials that assess the effects of DMT in healthy humans.

| Trial Identifier | Sponsor | Study Type | DMT Administration | Key Outcomes | References |
|---|---|---|---|---|---|
| NCT05559931 | Algernon Pharma | Phase 1 double-blind randomized control trial | intravenous infusion over six hours with loading bolus and infusion | overall safety, ECG, vitals, physical examination, lab values, injection site reactions, C-SSRS, psychotic symptoms, 5-HT toxicity syndrome, adverse events | [110] |
| NCT05384678 | University Hospital, Basel, Switzerland | Phase 1 double-blind crossover study | intravenous infusion with varying DMT doses and placebo | altered states of consciousness profiles (5D-ASC), subjective effect ratings | [111] |
| NCT05695495 | University Hospital, Basel, Switzerland | Phase 1 randomized trial | intravenous DMT bolus doses and placebo | altered states of consciousness profiles (OAV), subjective effect ratings | [112] |
| NCT04353024 | University Hospital, Basel, Switzerland | Double-blind crossover study | four different DMT doses and placebo | altered states of consciousness, subjective effect ratings, mood ratings, mystical-type experiences, autonomic effects, plasma levels, oxytocin levels, renal clearance, adverse effects | [92] |
| NCT04716335 | Psychiatric University Hospital, Zurich | Phase 1 study | investigating neurodynamics of prosocial emotional processing following serotonergic stimulation with DMT and harmine | pharmacological EEG, behavioral outcome measures, biomarkers | [118] |
| NCT05901012 | Universidade Federal do Rio Grande do Norte | Phase 1 double-blind randomized control trial | inhalation of DMT at experimental and placebo doses | blood pressure, heart rate, respiratory rate, oxygen saturation | [113] |
| NCT05780216 | University Medical Center Freiburg | Early phase 1 study | functional magnetic resonance imaging (fMRI) assessment of brain connectivity before and after a meditation group retreat with ayahuasca or placebo | fMRI comparisons of brain connectivity, effects of DMT-enhanced mindfulness on the default mode network | [119] |
| NCT04711915 | Yale School of Medicine | Phase 1 non-randomized escalating dose study | intravenous DMT at two different doses | changes in blood pressure, heart rate, modified ASC scale readings, anxiety levels, drug reinforcing effects, overall tolerability | [120] |
| NCT05573568 | Biomind Labs Inc. and Universidade Federal do Rio Grande do Norte | Phase 1 non-randomized sequential assignment open-label study | two sessions of inhaled DMT with varying doses | clinical and psychiatric adverse events, blood pressure, heart rate, respiratory rate, oxygen saturation changes | [114] |

**Table 2.** *Cont.*

| Trial Identifier | Sponsor | Study Type | DMT Administration | Key Outcomes | References |
|---|---|---|---|---|---|
| NCT04640831 | Maastricht University, Maastricht, Netherlands; Goethe-Universität Frankfurt am Main, Institut Für Rechtsmedizin, Frankfurt am Main, Germany; 3GH Research PLC, Dublin, Ireland | Phase 1 in healthy volunteers | single doses of novel vaporized 5-MeO-DMT formulation (GH001) | dose-related increments in the intensity of psychedelic experience questionnaires ratings | [121] |

*5.3. DMT and 5-MeO-DMT as Emerging Treatments for Post-Traumatic Stress Disorder (PTSD) and Treatment-Resistant Depression*

The therapeutic potential of ayahuasca in treating a wide variety of mental illnesses, disorders, and neurological conditions is well known and reviewed well [5,122]. We will focus our discussion on DMT and its analogs in treatments for PTSD and major depressive disorder. Exposure to a single or a series of traumatic events can lead to post-traumatic stress disorder (PTSD), impairing multiple functional domains [123]. MDMA continues to show promise in treating PTSD [124] compared to selective serotonin reuptake inhibitors (SSRIs) [125] especially in veteran populations [126]. Preliminary evidence suggests that DMT may also be an effective treatment for PTSD in veterans. From 2017–2019, US veterans with PTSD were surveyed 30 days before and after undergoing a psychedelic treatment program using 5-MeO-DMT [127]. The survey included the PTSD checklist (PCL-5), Patient Health Questionnaire-2 (PHQ-2), Generalized Anxiety Disorder 2-item scale, Medical Outcomes Study-Cognitive Functioning (MOS-CF), Acceptance and Action Questionnaire II (AAQ-II), and The Depressive Symptom Index Suicidality Subscale. At the end of the treatment, the results showed an average reduction in PCL-5 of 34.2 (SD = 19.3), an average reduction in depression symptoms of 3.2 (SD = 1.8), an average reduction in the Generalized Anxiety Disorder 2-item scale of 2.9 (SD = 1.9, an average reduction in MOS-CF of −1.5 (SD = 1.0), and a reduction in The Depressive Symptom Index Suicidality Subscale of 2.3 (SD = 2.5) [127].

In a similar study from the same investigative team, Special Operations Forces Veterans with trauma exposure were treated with both ibogaine and 5-MeO-DMT [128]. The veterans (*n* = 86) were evaluated at pretreatment and then followed up at 1, 3, and 6 months. Younger age and higher initial levels of depression and anxiety were linked to better outcomes in life satisfaction, cognitive functioning, psychological flexibility, trauma symptoms, and acute effects like personal meaningfulness and spiritual significance at 1-month follow-up. A greater intensity in changes of consciousness (like personal meaningfulness, spiritual significance, and psychological insightfulness) correlated with significant improvements in mental health (like cognitive functioning and trauma symptoms) and psychosocial outcomes (like social relationships, life attitudes, behavioral changes, spirituality) over the 6-month period. The most important outcome was an increase in psychological flexibility (from baseline to 1-month follow-up) and mediated the relationship between intense changes in consciousness and reductions in trauma, depression, and anxiety symptoms at the 1-month follow-up. The study suggests that both the acute effects of the combined ibogaine and 5-MeO-DMT treatment and enhancements in psychological flexibility are critical in achieving positive outcomes, especially for younger veterans and those with greater symptom severity before treatment [128]. Although these results are promising, the limitations of a non-randomized, survey-based study of patient symptoms, such as unaccounted confounders and recall bias, can suggest only that further investigation into DMT for PTSD treatment is warranted.

Major depressive disorder (MDD) is one of the most common psychiatric illnesses worldwide, affecting between 12 and 20% of adults throughout their lifetime [129–132]. Annually, major depressive disorder affects 6% of adults [130]. Complications resulting from major depressive disorder are both extensive and cumbersome, including a reduced quality of life, mortality, cognitive impairment, diabetes mellitus, heart disease, and cancer [130,132]. The successful management of major depressive disorder can be difficult due to the high rates of recurrence and high variance in clinical course [130,133,134]. Up to 50% of patients with major depressive disorder fail to respond to standardized treatments [133,135]. The lack of remission is central to treatment-resistant depression (TRD), which can be considered part of major depressive disorder's heterogenous clinical spectrum, characterized by a distinct unresponsiveness to treatment [133]. Although treatment-resistant depression does not have a universal definition, major depressive disorder is often considered treatment-resistant depression when patients fail to reach remission after two trials of antidepressants from different classes [135].

Patients with treatment-resistant depression who wish to continue further trials often must use alternative treatments. Psychedelic therapy offers a novel class of therapeutics and an exciting burgeoning of evidence to patients who have struggled with traditional therapies. Ayahuasca shows significantly decreased MADRS scores in a trial against placebo at days 1, 2, and 7 post treatment [136,137]. However, this trial still had a notable placebo effect, highlighted by a 27% response rate at Day 7 [136]. Ayahuasca importantly follows the trend of psychedelics inducing rapid antidepressant effects, while traditional treatments can take at least 2 weeks of daily administration to induce clinically relevant effects [136,137]. Open-label studies show favorable results up to 21 days [3]. Two other double-blind RCTs with equal dosages of ayahuasca (1 mL/kg) found higher levels of BDNF [138] and an equalizing effect on cortisol levels between healthy and depressive volunteers, including placebo [139], both measured two days after administration and correlated with less depressive symptoms [3]. In an exploratory report of a Phase 1 non-randomized open-label escalating-dose study on DMT in humans (NCT04711915) [120], the results showcased that intravenous DMT is safe and tolerable for individuals with moderate to severe, treatment-resistant depression, with potential next-day antidepressant effects measured through a decrease in the HAMD-17 score (Hedge's g = 0.75, $p = 0.017$) [140]. A study at the Yale School of Medicine, with estimated completion by June 1, 2024, was designed to assess the physiological and psychological effects of DMT, administered intravenously at doses of 0.1 mg/kg and 0.3 mg/kg in a fixed order across two days. The primary outcomes include changes in blood pressure, heart rate, modified ASC scale readings, anxiety levels, drug-reinforcing effects, and overall tolerability. Although the phase 1 component of Reckweg et al.'s clinical trial [141] showed limited results, 7/8 patients with treatment-resistant depression successfully showed remission (MADRS ≤ 10) at day 7. Phase 2 implemented an individual dosing regimen rather than the single dose administration seen in phase 1, with three separate and increasing doses administered in the same day [141]. Finally, a phase 1 study with intravenously administered DMT fumarate (SPL026) showed great tolerability and safety over a broad dose range in preparation for phase 2 studies on major depressive disorder, with the most common adverse effects (7 out of $n = 32$) involving benign infusion or catheter site pain or reaction resolving by 1 h post administration [142]. Although all the adverse events in this study were considered mild to moderate, 78% (25 out of $n = 32$) of the participants, including placebo, experienced an adverse event. The complete results of the phase 2 trial have not yet been published, but preliminary results suggest that a remission rate of about 57% occurs at the three-month mark after a single dose of SPL026 [143]. These ayahuasca and DMT trials for treatment-resistant depression add to a promising future for psychedelics as therapeutic agents against major depressive disorder and treatment-resistant depression.

Overall, DMT, as a tryptamine compound, shows relatively mild adverse effects. The most common adverse events in an exploratory subject pool ($n ≤ 10$) were transient hypertension ($n = 2$) during the onset and after the resolution of effects, transient headache ($n = 2$) before the dose and during its effects, and anxiety ($n = 6$) before administration and during its effects. Additionally, none of the subjects experienced clinically relevant psychotic symptoms and passed a pre-dose baseline-matched Mini-Mental Status Exam prior to discharge [140]. Further studies documented the autonomic and adverse effects of DMT with bolus doses rapidly increasing blood pressure and heart rate, peaking within 2 min, while infusions caused only a mild, nonsignificant elevation, with vital signs normalizing within 15 min post infusion. Adverse effects, more pronounced with higher bolus doses compared to placebo, included heart palpitations, nausea, tiredness, uneasiness, and thirst. The most common post-session adverse effect was a headache, reported by seven participants. Additionally, one subject experienced several weeks of moderate to intense flashbacks starting two weeks post study. The study also noted two severe, unrelated events: a COVID infection with syncope and an elective surgery [92]. In comparison, first-line therapies for depression, generalized anxiety disorder, and post-traumatic stress disorder have side effects that can limit function and lead to patient distress. Two common

first-line therapies used for the treatment of each of these conditions include SSRIs and serotonin and norepinephrine reuptake inhibitors (SNRIs). For example, acute side effects such as nausea, diarrhea, sexual dysfunction, and weight change are routinely reported in commonly used SSRIs such as sertraline, fluoxetine, and paroxetine [144], and citalopram has an additional risk of QT interval elongation [145]. Additionally, SNRIs have a similar side effect profile with the added risk of clinically significant supine diastolic blood pressure increase [146]. These drugs are usually administered chronically compared to infrequent dosages of tryptamine compounds that have similar antidepressant activity when directly compared [147]. The use of tryptamine compounds could potentially decrease the chronic medication side effect burden.

### 5.4. Unlocking DMT's Promise in Treating Neurodegenerative Disorders

As research into the therapeutic uses for psychedelics has surged in recent years, their usefulness in the treatment of an array of psychiatric conditions has become relatively well documented. However, more of the recent literature has begun to explore the possible usefulness of psychedelics in the treatment of neurodegenerative diseases. Much of the postulated utility of psychedelics, and DMT specifically, stems from their neuroplastic capabilities, which could be used to target both the loss of neurons and the lack of progenitor cell proliferation in affected brain areas [75]. We briefly discuss three neurological conditions focusing on possible DMT as a therapeutic. An excellent review discusses psychedelics as a whole in potentially treating these states [18].

#### 5.4.1. Stroke

Rats treated with DMT following middle cerebral artery occlusion had decreased ischemic lesion volumes and improved motor function compared to controls. Rats administered DMT with a sigma-1 antagonist did not demonstrate these effects, suggesting that DMT exerts its effects by binding to the sigma-1 receptor [148]. Further, post-stroke rats exhibited a significantly improved recovery of neurological function following the administration of a sigma-1 agonist [149].

#### 5.4.2. Parkinson's Disease (PD)

As an animal model for PD, mice with 6-hydroxydopamine-induced lesions in the nigrostriatal system were treated with a sigma-1 agonist, PRE-084, for five weeks to examine the potential benefits of sigma-1 receptor agonism for the treatment of PD. Compared to the controls, the treated mice exhibited significant improvements in behavioral deficits and a partial restoration of dopamine levels in the dorsolateral striatum [150]. Agonists of 5-HT$_{1A}$ receptors in astrocytes promote antioxidative protection against dopaminergic neurodegeneration in Parkinsonian mice [151]. A known ligand of both the sigma-1 and 5-HT$_{1A}$ receptors, DMT could prove to be a disease-modifying agent in PD. Yet, no studies have been conducted on DMT directly in the treatment of Parkinsonian features; however, the already established neuroprotective potential of DMT lends it to be a promising area of future exploration.

#### 5.4.3. Alzheimer's Disease (AD)

Dimethyltryptamine (DMT) may exhibit promising therapeutic potential for Alzheimer's disease (AD), primarily by modulating brain-derived neurotrophic factor (BDNF) levels and mitigating neuroinflammatory pathways. Studies underscore the detrimental role of low BDNF levels in AD progression, including amyloid-beta (Aβ) accumulation, tau phosphorylation, neuroinflammation, and neuronal apoptosis [152]. Conversely, DMT enhances the BDNF expression, which is crucial for neural health and resilience against AD pathology. For instance, Neuropep-1, an agent increasing hippocampal and cortical BDNF levels, substantially improves spatial learning and memory deficits in AD mice models, highlighting the therapeutic significance of BDNF modulation in AD [153]. Low BDNF levels have been associated with characteristics of AD-like Aβ accumulation, tau

phosphorylation, neuroinflammation, and neuronal apoptosis [152]. Furthermore, DMT's role extends to promoting anti-inflammatory responses, as exemplified by the increased expression of anti-inflammatory cytokines like IL-10 and the proliferation of neural progenitor cells, crucial for neurogenesis and cognitive function restoration in AD. The transplantation of neural precursor cells overexpressing an IL-1 receptor antagonist, for example, rescue memory deficits and boost BDNF-expressing hippocampal cells, reinforcing the therapeutic potential of targeting neuroinflammatory pathways in AD [154]. Moreover, DMT, as a sigma-1 receptor agonist, effectively mitigates neuroinflammation and fosters neurogenesis, as evidenced by the reduction in Aβ1–42-induced astrogliosis in AD mouse models [76]. In a study on DMT's effects on stroke recovery in rats, Nardai et al. found that DMT also increased the expression of BDNF protein and other anti-inflammatory cytokines, like IL-10 [148]. DMT also induces the neural progenitor cell proliferation and neurogenesis of hippocampal cells through sigma-1 receptor activation (see Section 4 above), leading to improved spatial learning and memory in a mouse model [75]. These findings collectively advocate for DMT's multifaceted role in direct and indirect mechanisms that may enhance neuroprotective pathways, reduce neurodegeneration hallmarks, and improve cognitive functions. Although this is quite speculative and needs much more work, DMT may have a role in AD therapy. This concept has appeared in the literature and is not unique to this review [155].

## 6. DMT's Influence on Brain Network Reorganization and Consciousnesses: Insights from Psychedelic Neuroimaging

Considerable interest exists in fully understanding the impact of psychedelics on higher-order brain networks and consciousness. Previous ideas proposed that defects in the synthesis of endogenous psychedelic molecules were the etiology of psychotic disorders [156,157]. While this hypothesis has been largely abandoned, the great interest in the biological and biochemical mechanisms underlying psychedelic effects on neural networks and brain activity has remained. Functional magnetic resonance imaging (fMRI) studies using human participants with various psychedelic molecules have revolutionized our understanding of how serotonergic psychedelics impact the brain.

### 6.1. Unraveling the Impact of DMT on Brain Network Connectivity

Psychedelics like LSD, psilocybin, and DMT significantly alter brain network connectivity, leading to a state of increased dynamism and entropy in brain activity. These changes in network connectivity are critical for understanding the profound alterations in consciousness experienced during the psychedelic state. LSD [158] and psilocybin [159] also exhibit similar effects, reducing within-network functional connectivity while enhancing between-network functional connectivity. Additionally, the spatial overlap of LSD-induced brain function variations with serotonin receptor expressions highlights the importance of neurotransmitter systems in mediating these effects [160].

The effects of DMT's sister molecules, LSD and psilocybin, on participants compared to placebo, show a very profound alteration in functional organization, connectivity, and states of entropy within the brain. LSD affects the energy of brain states in a selective manner based on frequency, which suggests a higher degree of available brain states that are achievable within a psychedelic state [161]. Similarly, the functional connectivity of major brain structures was profoundly altered. The examination of association cortices and the thalamus leads to increased global connectivity under the influence of LSD in a pattern overlapping with serotonin receptor density, suggesting that LSD increases the global connectivity of major brain regions [162]. LSD also modifies thalamocortical interactions, a crucial pathway in sensory and cognitive processing [163,164]. In particular, LSD's action on the thalamus's ventral, pulvinar, and non-specific nuclei suggests alterations in the functional coupling with sensory cortices and associative cortex areas dense in 5-HT$_{2A}$ receptors [164]. This aligns with findings from LSD [163] and psilocybin [165], where alterations in thalamocortical connectivity are associated with the substances' psychedelic

effects (reviewed in [166]). Psilocybin increases the variability of signals within the anterior cingulate cortex and the hippocampus. Additionally, a blood oxygen level-dependent (BOLD) signal spectral analysis showed alterations in the default mode network, executive control network, and the dorsal attention network [167]. Interestingly LSD and placebo lead to a flattening of the principal cortical connectivity gradient suggesting that serotonergic psychedelics have the capacity to dissolve unimodal cortices and lead to more transmodal cortical connectivity. This may suggest that this integration can disrupt the segregation of concrete and abstract processing [168]. Psychedelics such as MDMA and psilocybin decrease the differentiation of higher brain networks, such as visual and sensorimotor resting state networks [169]. LSD and psilocybin also reduce the control energy needed for brain state transitions, leading to increased states of entropy within brain state dynamics, i.e., allowing for rapid transitions between states of activity [170]. Further, psilocybin therapy shows rapid antidepressant effects correlated with decreases in brain network modularity and increases in global network integration. This global increase in brain network integration post psilocybin therapy suggests a mechanism for its antidepressant action [159]. Similar effects are observed with LSD, where alterations in brain function and subjective experience are temporally specific, indicating a nuanced impact on brain connectivity and complexity [171]. These effects may be extended to DMT, given its close structural similarity to psilocybin and functional similarity to LSD.

Consistent with the results of other serotonergic psychedelics, DMT shows the capacity to increase global functional connectivity, the desegregation of neural networks, and a similar flattening of the principal cortical gradient [172]. Twenty participants were randomized to either 20 mg of DMT or 10 mg of sterile saline placebo IV and monitored over two sessions with resting-state fMRI and EEG. The findings indicated that DMT significantly increased global functional connectivity, particularly within the default mode network, frontoparietal network, salience network, and limbic network. On average, global functional connectivity increases across all brain regions, though specifically in the medial prefrontal cortex, dorsolateral prefrontal cortex, insula, and temporal-parietal junction. This increase in global functional connectivity correlates with the intensity of the experience. The disintegration of a few neural networks also occurs on dynamic resting-state functional connectivity with a decrease in synchronized activity within the default mode network and dorsal attention network. Additionally, the flattening of the principal cortical gradient was also observed to be consistent with previous fMRI findings of LSD and psilocybin. This ultimately indicates the capacity of DMT to decrease the functional separation between lower- and higher-level cortical processes. These network connectivity changes correlate with PET-derived 5-HT$_{2A}$ receptor maps. Overall, this study offers insights into DMT's predominant action on the brain's transmodal association pole, that area of the cortex with high amounts of 5-HT$_{2A}$ receptors that evolved to process advanced psychological phenomena [172].

Neuroimaging studies suggest that psychedelics including DMT have a powerful capacity to modify rigid and segregated cortical processes and allow for global integration. These effects could be responsible for their observed capacity to mediate stable alterations to personalities, allowing for a disintegration of ingrained mental patterns [173]. This propensity to create new neural connections could potentially have clinical utility in correcting cognitive distortions and pattern thinking.

### 6.2. Insights into Consciousness Alteration from DMT-Induced Changes in Brain Function and Perception

The nature of consciousness has remained elusive for centuries. No longer a philosophical construct, consciousness is now a neurobiological manifestation of brain network activity that can be studied with advanced brain scanning and neuroimaging techniques [174]. Studying altered states of consciousness with EEG, fMRI, positron emission tomography (PET), and functional near-infrared spectroscopy (fNIRS) can reveal disorders in consciousness [175]. The study of LSD, psilocybin, and DMT offers insights into altered states of

consciousness and their neurobiological foundations. For instance, as mentioned above, DMT's influence on EEG-measured neurophysiological properties correlates with specific changes in fMRI metrics, enriching our understanding of its neural basis [172]. Moreover, the resemblance between psychedelic-induced and psychotic states, particularly in thalamocortical connectivity patterns, provides a unique perspective on the neural basis of altered consciousness and perception [166].

All classic serotonergic psychedelic drugs such as LSD, psilocybin, and DMT produce an altered state of consciousness with acute effects, which are most often visual hallucinations. DMT is notably the most potent in its ability to alter consciousness and produce an immersive experience in another reality. Most intriguingly, the hyper-real otherworldly dimensions coupled with the perceived presence and interaction with entities is a unique and common experience under the influence of DMT.

A study that interviewed 36 individuals post DMT use, mostly Caucasian males, with a few females, focused on a breakthrough aspect of DMT use, termed 'the other' [176]. Users also describe experiences of nebulous worlds and intricate patterns, like a cosmic template or latticework, suggesting a fundamentally unique aspect of the 'reality'. The encounters with sentient entities were often described as benevolent, all-knowing, hyper-intelligent, and sometimes malicious. The nature of these entities varied, with some being familiar to the participants. The interactions with these entities included themes of teaching, guiding, and presenting insights. The possible neural mechanisms underlying DMT experiences might include the disintegration of the default mode network, and the resulting radical perceptual restructuring could be key factors. This process might involve the release of innate neural modules, including a 'hyperactive agency detection device', which may be conserved through evolution to identify and relate with human-like agents. This could explain the anthropomorphic nature of many DMT entities [176].

In an even larger study surveying 2561 participants, the respondents primarily experienced visual and extrasensory perceptions, often described as telepathic. The entities encountered were commonly labeled as 'beings, guides, spirits, aliens, or helpers'. Despite 41% of participants reporting emotions like fear or anxiety, most of the emotions experienced and attributed to the entities were 'love, kindness, and joy'. A significant majority of respondents believed these entities to be 'conscious, intelligent, and benevolent', existing in a different but 'real dimension of reality' and continuing to exist post encounter. Remarkably, 69% of participants received a message, and 19%, a prediction, about the future from these encounters. A striking change was observed in religious beliefs: more than half of the respondents who previously identified as atheists no longer did so after the experience. These encounters were profoundly impactful, being rated among the 'most meaningful, spiritual, and psychologically insightful experiences' of the respondents' lives, leading to lasting positive changes in their life satisfaction, purpose, and sense of meaning [177]. The DMT experience alters consciousness in ways that lead to interactions with hallucinatory beings that in most cases are positive, meaningful, and durable. This is the breakthrough experience that may have a role in the possible therapeutic efficacy of DMT in treating disorders of the mind. If guided with cognitive behavioral therapy before and after such experiences, DMT may facilitate the ability to change minds more readily.

Retrospective studies such as those described above are limited in several ways with subjective memory being of major concern. Studying these effects on participants in real time could perhaps provide controlled results. Recently, the extended use of DMT has shown the potential to perform controlled observations on its effects over longer periods of time [178]. This lengthened the typical transient DMT experience through a novel method of administration combining a bolus injection with a constant-rate infusion. A single-blind, placebo-controlled study with 11 healthy volunteers receiving up to four different doses of DMT over 30 min demonstrated safety with only transient anxiety and heart rate increases in the first 15 min. Plasma concentrations of DMT increased with the dose and showed a significant dose–effect relationship. Participants experienced substantial increases in intensity ratings with all tested doses, indicating the method's effectiveness in maintaining

stable drug effects. Participants' experiences of immersion and visual imagery largely followed subjective intensity scores, while experiences of entity encounters increased in later parts of higher doses [178]. This pilot study sets up the ability to monitor the DMT experience in a highly controlled fashion with dosing and time intervals. Further, these techniques may allow for more thorough neuroimaging that has the power to finely resolve the DMT-altered state of consciousness.

## 7. Future Directions and Challenges: Exploring the Potential of DMT and Other Psychedelics in Clinical Practice

Given that most of the clinical trials mentioned had results that were not reported, the future directions largely hinge upon the results of those papers. Notably, current studies with any psychedelic agent struggle with masking the acute effects from placebo [179]. Blinding presents a large potential confounder to any RCTs being performed using DMT or other psychedelics being studied [179]. NCT03861988, an RCT on the effects of intravenous ketamine under the masking of surgical anesthesia, found no difference in the primary outcome between ketamine and placebo, raising questions about the validity of the recent burgeoning of evidence in favor of ketamine and other psychoactive therapeutic agents [180], while a meta-analysis of perioperative ketamine for postoperative depression favored ketamine over placebo in the short term and long term [181]. However, these issues should not disqualify psychedelics as a potential therapeutic intervention. Newer trials have used independent raters and active placebos to partially account for participant expectancy and de-blinding [182]. Additionally, clinical recommendations still include ketamine as an adjunct or for use in treatment-resistant depression or in cases of depression with imminent suicidality [183–186]. Psilocybin is considered under similar guidelines [187] as a potential alternative treatment for depression. Although the research is not as developed as it is for ketamine, one clinical trial [188] comparing psilocybin to placebo found that one dose of psilocybin (0.215 mg/kg body weight) led to a decrease in MADRS depression scores compared to placebo for at least two weeks.

The comparisons we have made between DMT and psilocybin/ketamine underscore our justifications for looking into DMT as a practical therapeutic agent. DMT and psilocybin share a similar main mechanism of action, binding on 5-HT$_{2A}$ receptors [189–191], and DMT's exploratory studies showed subjective effects similar to a significant psychedelic experience in the higher dose group [92,140]. While these similarities serve as a baseline, the section before highlights how DMT's pharmacological differences from psilocybin and ketamine offer a perspective into how DMT in particular has potential outside of the traditional uses of other psychedelic therapeutics, mainly driven by its sigma-1 receptor activity [51].

Looking ahead into clinical practice, optimism about DMT can be taken from ketamine's trends after FDA approval. Ketamine's short duration of acute effects is a large driving force in its use as an antidepressant. Ketamine is already pursued as a treatment in ketamine clinics, with a market value of USD 3.1 billion in 2022 [192]. DMT could be used similarly to ketamine, using multiple shorter sessions rather than one long session compared to psilocybin treatments [193]. One trial that evaluated the clinical use of DMT with published results highlights that DMT, like ketamine, was administered in a typical hospital setting with minimal psychotherapy, and still resulted in decreased depression scores (NCT04711915) [140]. The study also compares the antidepressant effects of DMT and psilocybin, noting a smaller magnitude of effects with DMT, hinting at a need for a direct comparison to fully understand their differences and similarities. Intravenous DMT can be particularly useful as it provides clinicians more control over the psychedelic state (NCT04353024) [92]. Additionally, psilocybin sessions can be resource and manpower intensive [193], lending further credence to the idea that DMT can be a more practical intervention for both the patient and the clinician.

## 7. Conclusions

The discovery of the ability of psychedelics to induce rapid, sustained, functional, and structural neuroplastic changes has greatly expanded the neuropharmacologic repertoire in ways that are still being investigated. Their use in psychiatry marks a paradigm shift away from neurochemical-based interventions to rapidly induced changes to higher-order neural circuitry [100,194–197]. In addition to psychiatry, psychedelics may reshape how we think about the pharmacologic treatment of neurologic disorders as well [155]. In addition to BDNF, other psychedelics such as ibogaine increase the levels of glial cell-derived neurotrophic factor, GDNF, which has increased interest in investigating analogs for their potential to treat PD [198,199]. Additionally, interest is also increased in the potential for these serotonergic compounds and their potential utility for treating AD. Synapse loss in AD correlates with the extent of cognitive symptoms [200–202]. The ability of psychedelics to rapidly induce synaptogenesis makes these molecules attractive alternatives to currently existing therapies that are aimed at just symptom control and targeting pathologic protein aggregates.

Despite the rapid expansion of interest in psychedelics and their clinical utility, a relative paucity exists in pre-clinical studies that outline and investigate the specific actions and downstream mediators involved in the neuritogenic effect of individual molecules. This is likely in part related to their schedule 1 status and the relative inaccessibility of these molecules for investigation [203]. In this literature review, we must assume similarities in the mechanisms of action among DMT, other indoleamine-based psychedelics like psilocybin and LSD, and phenethylamines like DOI that target the 5-HT$_{2A}$ receptor. The pursuit of DMT's molecular mechanisms should persist in identifying the downstream mediators of neuritogenesis to verify their consistency across serotonergic psychedelic compounds.

**Author Contributions:** Conceptualization, F.A.C., P.B., R.I.K., R.E.M. and T.A.V.; writing—original draft preparation as follows, F.A.C.—introduction, DMT and neurological disorders, and DMT receptor interactions, P.B.—DMT molecular mechanisms in neurogenesis and neuritogenesis, Figure 1 design and legend, DMT in PTSD, future directions and challenges, and DMT and neural network connectivity, R.I.K.—DMT in animal models and clinical studies in humans, Tables 1 and 2, the section on major depressive disorder and treatment-resistant depression, and part of DMT metabolism, R.E.M.—DMT metabolism, and Table 1; writing—review and editing, T.A.V., P.B., R.I.K. and F.A.C.; project administration, F.A.C.; project supervision, T.A.V.; revisions after review, F.A.C., P.B., R.I.K., R.E.M. and T.A.V. All authors have read and agreed to the published version of the manuscript.

**Funding:** This research received no external funding.

**Acknowledgments:** Many references were searched for and retrieved using PubMed and Semantic Scholar. We acknowledge the use of artificial intelligence (AI) in the preparation of the work. For example, ReseachRabbit.ai was used to find the initial literature, organizing it into themes and common ideas. Elicit.com (accessed on 30 October 2023) was used with specific research questions to target unique areas that needed precise evidence. GPT-3.5 and GPT-4 (OpenAI) were used to construct initial outlines of the manuscript that we subsequently modified extensively with our structure and plan. GPT-4 with and without plug-ins (Consensus Search, ScholarAI, and PubMed search) was also used to find references to support ideas not readily found using conventional methods. GPT-4 was also used to construct subheading statements and to synthesize rough drafts of some review sections, namely 6.2. These rough drafts were then rewritten to include the complete ideas needed with appropriate citations. GPT-3.5 and GPT-4 were also used to periodically check grammar and word usage.

**Conflicts of Interest:** The authors declare no conflicts of interest.

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
