# Peer review of "The Clinical Potential of Dimethyltryptamine: Breakthroughs into the Other Side of Mental Illness, Neurodegeneration, and Consciousness"

_psychoactives, doi:10.3390/psychoactives3010007_

Round 1
Reviewer 1 Report (Previous Reviewer 1)
Comments and Suggestions for Authors
After complete round of review, I am satisfied, the review can be accepted for publication.
Author Response
see attached file

Reviewer 2 Report (Previous Reviewer 2)
Comments and Suggestions for Authors
The authors have adequately addressed all of my concerns.
On a minor note, I think that calling DMT a neuromodulator would still be controversial, due to the lack of evidence suggesting its vesicular release via presynaptic terminals. I'd call it an endogenous ligand capable of neuromodulation, but I'd let the authors and editors decide what would be the appropriate definition.
Comments on the Quality of English LanguageMinor editing for tense agreement may be necessary.
Author Response
see attached file

Reviewer 3 Report (Previous Reviewer 3)
Comments and Suggestions for Authors
We appreciate this reviewer’ comments. The attention to detail has significantly improved the manuscript. The reviewer appears to have thoroughly examined each reference, which is admirable.
References are crucial in a review.
However, in this detailed analysis, several key points were missed.
I don’t know how the authors can support this statement.
We also note that the extensive use of merely stating manuscript lines and not specific ideas has made addressing these comments rather difficult. We suggest an approach that includes more specifics be used in the future. The line numbers rarely matched the statements.
Reviewers receive manuscripts with line numbers to ease the identification of sentences to be modified. My suggestion is to download the submitted manuscript with line numbers in the future.
Nonetheless, we have addressed all the requested changes and changed those that were in error or needed clarification.
Most concerns have been addressed. I’ll explain more clearly those minor comments that were not adequately addressed.
References 36-39 are mistakenly cited at line 203: the sentence in the revised manuscript is: “For example, psilocybin causes rapid, lasting changes in dendrite architecture in vivo after a single dose in mice [22,37–40,70]”. References 37-40, which were 36-39 in the previous version, do not report treatments with psilocybin in mice.
Lines 213-215: the reference for this effect is missing. You answered: “We referenced this paper (citation 55) for this effect: Ly, C.; Greb, A.C.; Cameron, L.P.; Wong, J.M.; Barragan, E.V.; Wilson, P.C.; Burbach, K.F.; Zarandi, S.S.; Sood, A.; Paddy, M.R.; et al. Psychedelics Promote Structural and Functional Neural Plasticity. Cell Rep. 2018, 23, 3170–3182, doi:10.1016/j.celrep.2018.05.022. This work demonstrates the neuritogenic effects of DMT. The data are quite clear in showing dendritic spine formation (Fig. 2,I therein).” The reference is fine, but in this case, you should move it after the statement “Even at sub-hallucinogenic doses, similar effects were observed.”, or repeat it at that point.
Lines 254-256: the effect is unclear: how can an agonist reverse constitutive activation cannot be easily grasped, my suggestion is either to remove or to expand. This comment is about: “Constitutive activation of 5-HT6 through its interaction with Cdk5 leads to neurite growth, which was abolished through the administration of a 5-HT6 receptor agonist [80].” Your sentence is at variance with what was reported in the cited paper, which described that the abolishment was caused by antagonist treatment.
I still disagree with some interpretations, but I accept that the authors wish to maintain them.
Author Response
see attached file

This manuscript is a resubmission of an earlier submission. The following is a list of the peer review reports and author responses from that submission.
Round 1
Reviewer 1 Report
Comments and Suggestions for Authors
I have found this review very informative and comprehensive. It contained all the necessary information that is required. All the possible mechanisms and interactions of the DMT have been elaborated. Except for a few sections that are more detailed and distort the reader's attention such as., Section 7: future directions and challenges, should be summarized and the extra information should be deleted. Additionally, a little modification of the abstract is needed. There should be more clarity of objectives and novelty of this review.
Reviewer 2 Report
Comments and Suggestions for Authors
This is an interesting review discussing the therapeutic potential of dimethyltryptamine (DMT) for a range of neurological disorders. The study details mechanistic insights and therapeutic efficacy of DMT and its analogs, gained from animal studies and clinical trials alike. However, the review fails to discuss the following major, critical aspects of DMT use as a therapeutic agent.
1) The authors refer to endogenous DMT as a neurotransmitter at several places in the manuscript. This seems speculative, as to my knowledge, there’s no study suggesting an activity-dependent release of DMT from axonal presynaptic terminals. Although authors cite a study that shows DMT binding to SERT and VMAT-2, this evidence is still indirect. The authors should not refer to endogenous DMT as a neurotransmitter or discuss the lack of supporting evidence.
2) The author’s claim that DMT-induced “neuroplasticity” underlies the therapeutic efficacy of DMT. Neuroplasticity is a broad term that includes structural plasticity at neurites and spines, along with homeostatic and Hebbian plasticity paradigms that are predominantly expressed at pre and postsynaptic sites. The authors should not be generalizing these plasticity mechanisms as they involve district underlying molecular processes. While DMT treatment has shown to cause neurite outgrowth and increased spine density in rodents, there is less evidence suggesting its involvement in other plasticity mechanisms. Moreover, it is unclear if these seemingly neuroplastic effects of DMT administration are due to its direct action on 5-HT receptors, or an indirect effect like increased glutamatergic tone or excitation/inhibition ratio resulting from altered neuromodulation. I’d encourage authors to appropriately revise the manuscript.
3) The authors identify serotonin receptors as one of the key downstream targets of DMT. Serotonergic modulation may confer DMT’s therapeutic efficacy in treating neuropsychiatric conditions, while Sigma-1 and Trk-B activation through DMT may be more relevant for its neuroprotective actions. Authors need to discuss serotonergic modulation by DMT as a potential mechanism when treating neuropsychiatric conditions. Authors may also compare DMT’s efficacy with currently used SSRIs for the management of neuropsychiatric conditions like anxiety/depressive disorders.
4) DMT is a potent hallucinogen, but even at sub-hallucinogenic doses, it can cause adverse effects like increased heart rate/blood pressure, confusion, dizziness, loss of muscle coordination, etc. The manuscript lacks a detailed discussion on the adverse effects of DMT when treating different neurological conditions, especially when the clinical trials included in the study evaluated them as a key outcome. Additionally, authors should compare the adverse effects of DMT with those associated with current therapeutic options for neuropsychiatric and neurodegenerative disorders.
Comments on the Quality of English LanguageThe manuscript is written well. Minor editing may improve the manuscript, but not necessary.
Reviewer 3 Report
Comments and Suggestions for Authors
The manuscript entitled “The clinical potential of dimethyltryptamine: breakthroughs into the other side of mental illness, neurodegeneration, and consciousness” by Colosimo et al is a review on potential therapeutic applications of natural compounds in ayahuasca.
The topic is interesting. There are some concerns that the authors should address.
Line 15: “neuronal neuroplasticity” is redundant.
In the abstract, the proposed mechanisms (lines 21-23) are reported after the applications, which is confusing.
The abbreviation PDSD should be expanded in the abstract.
Lines 50-51: please rephrase the sentence, the similarities are between DMT effects and the positive symptoms, not between DMT itself.
Line 102: “spine” should be “spinal cord”. Same at line 343.
Lines 131-132: “injection” should be more specific by describing the administration route.
Lines 140-142: a MAO inhibitor is supposed to reduce DMT-NO production, isn’t it?
A figure displaying the mentioned structures can be helpful.
Lines 160-161: the verb is missing.
Citations 52 and 53 are mistakenly cited in the list.
References 36-39 are mistakenly cited at line 203; citations 56-59 and 61-67 are missing in the text.
Lines 213-215: the reference for this effect is missing.
Lines 254-256: the effect is unclear: how can an agonist reverse constitutive activation cannot be easily grasped, my suggestion is either to remove or to expand.
Line 264: by “sites” are 5HT6 receptors meant? If this is the case, please mention them.
Lines 266-267: a rise that is not significant is not relevant, it may belong to random noise.
Section at lines 280-287 is an introduction to BDNF that should be placed at the beginning of this topic and shortened.
Line 305: no evidence is available about the proposed mechanism.
Figure 1: the mechanism mentioned in lines 312-313 (sigma and BDNF) is not demonstrated by cited papers, thus it should be removed. Also 320-21.
The section at lines 334-357 is unclear and should be rephrased, especially 339-344.
Line 364: “DMT effects on anxiety” would be more appropriate than anxiolytic, since it is reported to be anxiogenic.
Line 439: is the outcome available, since it has been completed in 2021?
Lines 441-451: a clinical trial is described which is not cited: the citation should be introduced.
Lines 472-475: the statement should be downplayed as very few results are available to support it.
Lines 513-516: the sentence is unclear.
Section starting at line 517: some general references could be removed. Reference 3 at line 541 should be replaced with citations reporting the specific trials which observed the effect. Line 541: the verb is missing.
Lines 545-549: the study did not investigate DMT, but the oxide; please replace.
Line 578: 6-hydroxydopamine-induced lesions.
Section 5.4.3: this interpretation seems far-fetched, I suggest a more cautious approach.
Section 6.1: at lines 611-615, the first sentence is redundant. The section is too long, it could be shortened and summarized, also limiting references to the most important studies.
Section 6.2: Likewise, this entire section is based on reporting the results of three studies. In my opinion, it should be summarized and reduced.
Line 757: reference 169 is about SUD, not PD.
Lines759-762: there is no evidence for this statement.
I am unable to access the dynamedex references.
Please check that abbreviations are spelled out at first use and also use them sparingly because they are very confusing to readers: those that are mentioned only a couple of times could be avoided.
If AI tools have been used, this should be acknowledged in the Acknowledgements section, with a description of the purpose.
